# A changing thermal regime revealed from shallow to deep basalt source melting in the Moon

Yash Srivastava[1,2], Amit Basu Sarbadhikari [1] ✉, James M. D. Day [3], Akira Yamaguchi[4] & Atsushi Takenouchi[4,5]

Sample return missions have provided the basis for understanding the thermochemical evolution of the Moon. Mare basalt sources are likely to have originated from partial melting of lunar magma ocean cumulates after solidification from an initially molten state. Some of the Apollo mare basalts show evidence for the presence in their source of a late-stage radiogenic heat-producing incompatible element-rich layer, known for its enrichment in potassium, rare-earth elements, and phosphorus (KREEP). Here we show the most depleted lunar meteorite, Asuka-881757, and associated mare basalts, represent ancient (~3.9 Ga) partial melts of KREEP-free Fe-rich mantle. Petrological modeling demonstrates that these basalts were generated at lower temperatures and shallower depths than typical Apollo mare basalts. Calculated mantle potential temperatures of these rocks suggest a relatively cooler mantle source and lower surface heat flow than those associated with later-erupted mare basalts, suggesting a fundamental shift in melting regime in the Moon from ~3.9 to ~3.3 Ga.

There is a broad consensus that most of the Apollo returned mare basalts contain variable quantities of material derived from a KREEP (potassium, rare earth element and phosphorus) component mixed into their mantle source(s), possibly due to mantle overturn after extensive lunar magma ocean (LMO) crystallization[1], widely known as 'urKREEP'[2]. The 'KREEPy-ness' of these basalts are supported by high abundances of other incompatible trace elements (ITE). A complementary argument posits that some lunar basalts are of KREEP-free origin with ITE enrichment produced purely through simple magmatic processes such as low-degree melting of depleted lunar mantle followed by extensive fractional crystallization; a mechanism distinct from urKREEP addition[3–6]. KREEP-free mantle sources are considered to have low initial $^{87}Sr/^{86}Sr$ (≤0.700) and high $^{143}Nd/^{144}Nd$ or positive $\varepsilon_{Nd}$, in contrast to KREEP-rich materials which have high initial $^{87}Sr/^{86}Sr$ (>0.701) and low $^{143}Nd/^{144}Nd$ or negative $\varepsilon_{Nd}$.

While ambiguities related to the exact origin of KREEP components exist for some mare basalts, the search for extremely depleted mare basaltic samples remains critical for characterizing the earliest (before mantle overturn) LMO cumulates, which are predicted to be relatively devoid of the ITE and heat-producing elements. Study of these samples would enable elucidation of the early thermal and chemical state of the lunar interior. Previously identified KREEP-free lunar basalts (NWA 032[3]; LAP basalts[4]; Chang'E 5 basalts[6]) are typically low in $TiO_2$ (0.45–5.70 wt.%) and are Fe-rich (Mg# [molar Mg/(Mg+Fe) × 100] 32–49; Fig. 1), with characteristically low initial $^{87}Sr/^{86}Sr$ (0.699–0.700) and positive $\varepsilon_{Nd}$ (+0.8 to +7.0) indicating mantle source depletion (Fig. 2).

In this work, we investigate the petrogenesis of the 'YAMM' (Y-793169, A-881757, MIL 05035, MET 01210) lunar meteorites, with special emphasis on Asuka-881757 (A-881757) to understand ancient lunar

[1]Physical Research Laboratory, Ahmedabad 380009, India. [2]Indian Institute of Technology Gandhinagar, Gujarat 382355, India. [3]Scripps Institution of Oceanography, University of California San Diego, La Jolla, CA 92093-0244, USA. [4]National Institute of Polar Research (NIPR), Tokyo 190-8518, Japan. [5]The Kyoto University Museum, Kyoto Universiti, Kyoto 606-8501, Japan. ✉e-mail: amitbs@prl.res.in

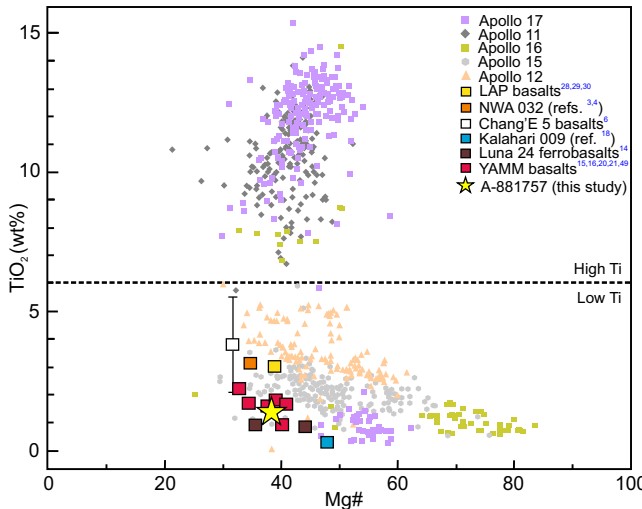

**Fig. 1 | Bulk major-element composition comparison among A-881757, YAMM and other lunar basalts.** $TiO_2$ plotted against bulk Mg# (molar Mg/(Mg+Fe) × 100), showing YAMM basalts are the low-Ti and most Fe-rich variety. Compositions of Apollo basalts are taken from the ApolloBasalt DB(v2) database[72].

volcanism. These basaltic meteorites are likely launch-paired as suggested by their similar mineral assemblage, pyroxene crystallization trends, REE abundances, crystallization ages (~3.8–3.9 Ga[7–9]), and low U/Pb, Rb/Sr, and high Sm/Nd[10]. The YAMM meteorites span crystallization ages broadly coinciding with the putative late heavy bombardment (LHB)[11] that pre-dates Apollo (~3.8–3.1 Ga) and Chang'E 5 (~2.0 Ga) mare basalts. The YAMM meteorites are Fe-rich (Mg# 32–41; Fig. 1) with low bulk REE and other ITE (e.g., 0.48 ppm Th). The low Rb/Sr ($^{87}Sr/^{86}Sr_i = 0.69908–0.69910$), high Sm/Nd ($\varepsilon_{Nd} = +7.2$ to $+7.4$), and unusually low U-Pb ($\mu = 9–20$) of their sources suggest a KREEP-free source distinct from the Procellarum KREEP Terrane[12] (PKT) basalts (Fig. 2).

## Results and discussion

### Petrology and geochemistry of YAMM basalts

Asuka-881757 is a coarse-grained (up to 8 mm), holocrystalline, unbrecciated mare basalt with pyroxene and plagioclase as primary constituents (Supplementary Figs. 1 and 2), consistent with other YAMM basalts. Coarse-grained basalts with gabbroic textures are rare amongst the returned samples[10,13]. Sample A-881757 pyroxenes are Fe-rich and cover a wide spectrum of compositions (Supplementary Text 1, Supplementary Fig. 3 and Supplementary Data 1), consistent with other YAMM and Luna 24 ferrobasalts[14] (Supplementary Text 2). The compositional zoning in maskelynitized plagioclase grains range from $An_{96}$ to $An_{87}$ with rims close to late-stage crystallizing phases reaching $An_{75}$ (Supplementary Fig. 4), following a typical fractionation trend shown by other low-Ti lunar basalts. Unlike most Apollo, Luna, and Chang'E basalts, forsteritic olivine is absent in A-881757.

Asuka-881757 is a very low-Ti, low-Al, and low-K basaltic rock (Supplementary Data 2)[15,16]. It consists of low Ti/Sm (0.96), low Th/Sm (0.16) and low Th/Hf (0.24). Further, the CI-normalized (cn) LREE-depleted values ($[La/Sm]_{cn} = 0.68$, $[La/Lu]_{cn} = 0.67$ and $[La/Yb]_{cn} = 1.05$) and a relatively flat HREE ratio ($[Tb/Yb]_{cn} = 1.04$) are consistent with other YAMM basalts (Fig. 3 and Supplementary Data 2). A similar LREE depleted pattern is also observed for the low-Ti Luna 24 ferrobasalts ($[La/Sm]_{cn} = 0.43$)[17], and Kalahari 009 ($[La/Sm]_{cn} = 0.91$)[18]. These patterns are distinct from the KREEP basalts (high Th/Sm = 0.45; $[La/Sm]_{cn} = 1.46$ and $[La/Yb]_{cn} = 2.13$)[2] or Apollo basaltic rocks with KREEP components ($(La/Sm)_{cn} = 1.32–1.50$)[19] (Fig. 3). Unlike most Apollo mare basalts, A-881757 and other YAMM

members, Kalahari 009 and Luna 24 ferrobasalts do not possess a negative Eu-anomaly (Fig. 3).

The most LREE-depleted YAMM meteorites suggest limited fractionation from a primitive partial melt[20]. Notably, the relative REE abundances of MIL 05035 ($[La/Sm]_{cn} = 0.55–0.63$ and $[La/Lu]_{cn} = 0.41–0.48$)[20,21] is the lowest in the YAMM meteorites, suggesting MIL 05035 represents the most primitive melt while partial melting links the clan members (Supplementary Figs. 5 and 6). Using exchange coefficient, $K_{D,Fe-Mg}^{Pyx-melt} = 0.28$[22,23], the melt in equilibrium with the most magnesian pyroxene (Mg# 54 in A-881757 and MIL 05035, and Mg# 61 in Y-793169) would have Mg# 25 and 30, respectively. The bulk compositions of A-881757 and MIL 05035 are however more Mg-rich (Mg# 34–38 in A-881757, Mg# 37–40 in MIL 05035). The difference between the measured and calculated compositions of Mg-rich pyroxenes in these samples indicates partial pyroxene accumulation in the whole-rock, consistent with samples approaching a parental melt composition (Supplementary Text 3 and Supplementary Fig. 7).

Strontium-neodymium isotope systematics indicate a distinct depleted mantle source produced the YAMM meteorites ($^{87}Rb/^{86}Sr = 0.019$; $^{147}Sm/^{144}Nd = 0.24–0.31$), which are the most depleted among mare basalts (Fig. 2b and Supplementary Fig. 8). Along with the lack of Eu anomalies, Sr-Nd isotope systematics indicate that the YAMM and other KREEP-free basalts (Luna 24 ferrobasalts, Kalahari 009) have sources formed prior to large-scale plagioclase separation. Sample analysis and experiments reveal that a stratified relatively Fe-rich upper mantle is viable from the low-pressure origin of Luna 24 ferrobasalts[24]. Collectively, these results demonstrate that the YAMM meteorites along with Kalahari 009 and Luna 24 ferrobasalts originated from a distinct depleted, Fe-rich and plagioclase-saturated lunar mantle that is free of urKREEP.

A critical aspect of the KREEP-free basalts is to determine the degree of melting at the YAMM basalts' source region in the backdrop of LMO scenario[3,5,25,26]. Using trace-element modeling (Methods, Supplementary Text 3 and 5), the YAMM basalts are consistent with forming from ~3–6% partial melting of the LMO cumulates at ~75–80 percent solid (PCS) containing ~1% trapped interstitial residual liquid (TIRL) (Supplementary Fig. 9 and Supplementary Data 3). The other KREEP-free lunar basalts, e.g., Kalahari 009 and Luna 24 ferrobasalts underwent 7–9% and 3–6% partial melting, respectively, of their mantle sources (Supplementary Figs. 9 and 14). Similarly, the Apollo mare basalts also underwent ~1–9% partial melting of their respective mantle source[5,25,27] (Supplementary Fig. 10).

Contextually, the younger (~2.9 Ga) LAP and NWA 032 basalts are also KREEP-free and, therefore, merit discussion. Despite being unrelated to early formed KREEP reservoirs[3,4], LAP and NWA 032 basalts have elevated ITE (~40–50 × CI of LREE and ~30–45 × CI of HREE) abundances, prominent Eu anomaly and LREE-enriched ($[La/Sm]_{cn} = ~1.10–1.17$) signatures (Fig. 3), quite similar to that of the Apollo basalts[19]. Because NWA 032 and the LAP basalts have low initial $^{87}Sr/^{86}Sr$ (0.699–0.700) and positive $\varepsilon_{Nd}$ (+3.0 to +9.7), even a limited (<0.5%) urKREEP contribution would significantly shift their Sr-Nd isotopic composition[3,4] towards high $^{87}Rb/^{86}Sr$ (>0.19) and low $^{147}Sm/^{144}Nd$ (<0.173) ratios (Supplementary Fig. 8). It was postulated that the LAP basalts were either extensively fractionated[28–30] or formed by low-degree of partial melting (0.7–1.5%) from an Fe-rich cumulate mantle[4] without urKREEP signatures[31]. The elevated ITE abundances relative to their low Mg# and previously suggested Apollo 12 like parent composition[28–30] indicate LAP and NWA 032 basalts to be fractionation products. However, the chemical modeling suggests that the REE composition of LAP and NWA 032 basalts can be reproduced by a small degree of partial melting[4,31]. While the petrogenesis of these rocks remains controversial, we calculated their estimated formation pressure-temperature (P-T) using whole-rock compositions, considering that these basalts represent melt composition based on the latest petrological experiments[31] (Supplementary Text 6).

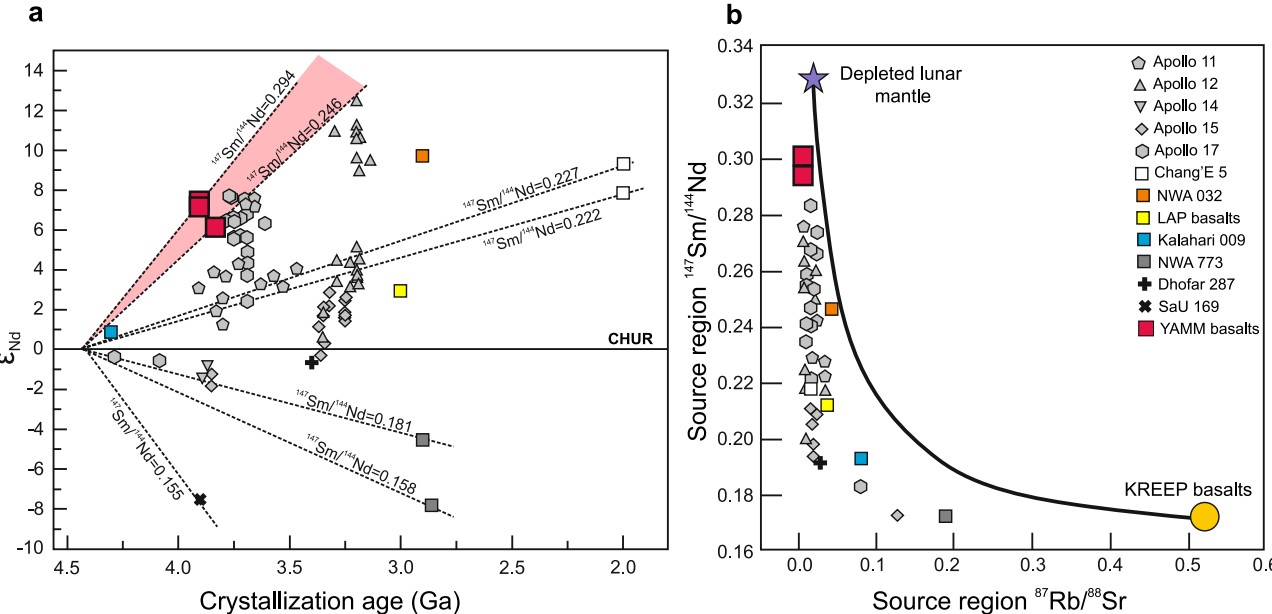

**Fig. 2 | Sm-Nd and Rb-Sr source isotopic characteristics of YAMM meteorites and other lunar basalts. a** Plot of age (Ga, billion years) against $\varepsilon_{Nd}$ composition showing the depleted nature of YAMM meteorites in comparison to other lunar meteorites and Apollo mare basalts. $\varepsilon_{Nd}(t) = (({}^{143}Nd/{}^{144}Nd)sample(t)/({}^{143}Nd/{}^{144}Nd)_{CHUR} - 1) \times 10,000$, where $({}^{143}Nd/{}^{144}Nd)sample(t)$ and $({}^{143}Nd/{}^{144}Nd)_{CHUR}$ are the Nd isotopic compositions of sample and Chondritic Uniform Reservoir (CHUR) at time ($t$) of crystallization of the rock, respectively. **b** Calculated present day values of ${}^{147}Sm/{}^{144}Nd$ plotted against the present day ${}^{87}Rb/{}^{86}Sr$ for the source

regions of YAMM and other lunar mare basalts (cf. ref. 4) using a model where the Moon differentiates at 4558 Ma with an initial ${}^{87}Sr/{}^{86}Sr$ of LUNI = 0.69903 (ref. 73). The ${}^{147}Sm/{}^{144}Nd$ of basalt source regions are calculated assuming a two-stage model where the Moon follows a chondritic path until differentiation occurs at 4.42 Ga, at which time mare basalt source regions were formed. Data Sources: YAMM meteorites (A-881757, ref. 7; Y-793169, ref. 8; MIL 05035, ref. 9), Kalahari 009 (ref. 74), Chang'E 5 basalts[6] and other Apollo mare basalts and meteorites (ref. 4 and references therein).

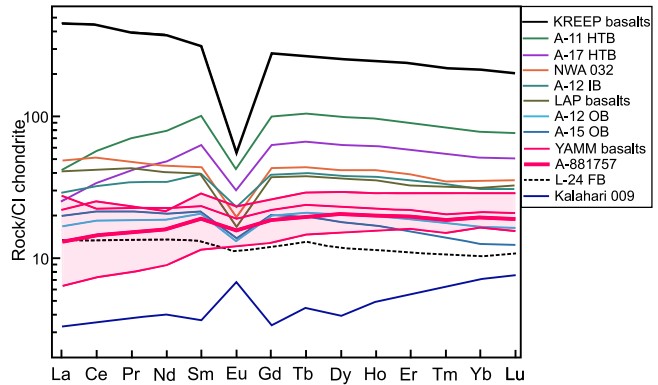

**Fig. 3 | Measured REE abundance of A-881757 compared with other YAMM and lunar basalts.** Chondrite-normalized rare earth element plots for YAMM (magenta), Kalahari 009 (blue), Luna 24 ferrobasalts (FB; dashed line), NWA 032 (orange), LAP basalts (olive green), along with Apollo 12 and Apollo 15 olivine basalts, Apollo 12 ilmenite basalts, Apollo 11 and Apollo 17 high-Ti basalts and KREEP basalts. YAMM basalts along with Luna 24 ferrobasalts and Kalahari 009 show depleted REE pattern while there is a lack of prominent Eu anomaly in YAMM and Luna 24 ferrobasalts, which are distinct from Apollo mare and KREEP basalts. Data Sources: A-881757 (this study); Y-793169 (ref. 16); MIL 05035 (ref. 20); MET 01210 (ref. 75); Luna 24 ferrobasalts[14]; Kalahari 009 (ref. 18); Apollo and KREEP basalts[19]. Abbreviations are A-11 HTB: Apollo 11 high-Ti basalts; A-17 HTB: Apollo 17 high-Ti basalts; A-12 IB: Apollo 12 ilmenite basalts; A-12 OB: Apollo 12 olivine basalts; A-15 OB: Apollo 15 olivine basalts; L-24 FB: Luna 24 ferrobasalts.

## Calculation of formation P-T

We use pMELTS[32,33] model calculations and traditional thermobarometric approaches[34,35] to estimate the formation $P$-$T$ of A-881757 (YAMM), KREEP-free samples, and Apollo basalt parental melts (Methods). The formation of A-881757 and YAMM basalts initiated at 0.3–0.8 GPa $P$ and 1100–1190 °C $T$. By comparison, the calculated $P$-$T$

of the oldest mare basalt Kalahari 009 is 0.7–1.0 GPa, and 1195–1235 °C (Fig. 4a and Supplementary Fig. 12). Collectively, formation of the KREEP-free mare basalts in the lunar mantle range between 0.3–1.0 GPa (~60–200 km) and 1100–1235 °C (Fig. 4a and Supplementary Data 4). The estimated $P$-$T$ conditions are significantly lower when compared to that obtained for the melt compositions of the Apollo mare basalts (1310–1410 °C at 0.9–1.3 GPa; Supplementary Data 4, Supplementary Text 4 and 7). Our calculated $P$-$T$ values for the Apollo samples are similar to experimentally derived results[31,36] (Supplementary Text 7). The experimentally derived $P$-$T$ conditions of the Apollo picritic glasses are even higher (1430–1560 °C at 1.3–2.5 GPa; refs. 31, 36 and references therein) than the lunar basalts. Among the returned lunar samples, only Luna 24 ferrobasalts show a similar $P$-$T$ range (0.4–0.5 GPa and 1180 °C)[24]. The lower estimated temperatures can be attributed to the lower Mg# (48–32) than the Apollo mare basalts, although the bulk composition of the KREEP-free rocks suggest that they were in equilibrium with their lunar mantle sources. To understand this disparity in the formation conditions between the KREEP-free basalts and the Apollo mare basalts, we further calculate mantle potential temperature ($T_p$) to examine likely thermal conditions.

## Mantle potential temperature and surface heat flux

Our result indicates that the KREEP-free basalts were sourced from a cooler lunar mantle than the mantle source of Apollo mare basalts. Estimates of average $T_p$, considering ~1–9% partial melting (Supplementary Text 4 and Supplementary Fig. 10), correcting for the average $P$-$T$ using lunar mantle adiabatic gradient, and correcting for the effect of latent heat of fusion on $T$ (Methods), ranges from 1110–1250 °C for the KREEP-free basalts (Supplementary Data 4). The highest $T_p$ of 1210–1250 °C is yielded by the oldest (~4.36 Ga) Kalahari 009, followed by 1110–1200 °C by YAMM basalts (~3.9 Ga), and 1160–1170 °C by Luna 24 ferrobasalts. The youngest NWA 032 and LAP basalts yield 1150–1190 °C $T_p$. These values are, however, lower than that of our

calculations (1295–1385 °C for Apollo basalts) and the results obtained for picrites (1350–1450 °C) from previous studies[36,37]. Even when the error (up to ±50 °C; Supplementary Data 4) is considered in the calculated $T_p$, there remains a significant difference in $T_p$ (100–200 °C) between the generally older KREEP-free and younger PKT samples. Such differences can be explained in terms of the distinct evolution of the KREEP-free basalts in comparison to the later mare basalts, highlighting the regional differences between PKT and non-PKT regions.

Lower $T_p$ and shallower depth of formation of the KREEP-free basalts in comparison to the Apollo basalts imply lower surface heat flow (56–63 mW/m²) possibly due to KREEP-free magmatism compared to that at the PKT region (64–67 mW/m²), assuming magma stalling at 40 km depth for both cases (Fig. 4b, Supplementary Data 4 and Supplementary Fig. 13). In all likelihood, the PKT mare basalts were stalled at greater depths beneath the surface[31,36] while the YAMM basalts were emplaced at shallower depths, which influences surface heat flow. The high heat flow at the PKT region was plausibly controlled by mantle overturn induced KREEP accumulation at source, or by some other process. The low $T_p$ obtained for the non-PKT samples even at 3.9 Ga can explain the likely causes of the lower mare basalt fill at the South Pole Aitken basin and Feldspathic Highland Terrain than the PKT region, where $T_p$ was elevated even up to 3.3 Ga.

Calculated high $T_p$ and elevated surface heat flow values obtained from Apollo mare basalts indicate they reflect a localized thermal anomaly possibly near the PKT region. Colder $T_p$ and lower heat flows of KREEP-free regions, having low-Th (YAMM, Kalahari 009 and Luna 24 ferrobasalts), obtained from samples of varying ages and localities may likely provide higher fidelity information on the global thermal regime. Our results also support the long-standing idea that the lunar mantle was thermally heterogeneous at various scales[36,38,39]. The lower degree of melting and relatively Fe-rich source of KREEP-free rocks in comparison to the PKT mare basalts may indicate that a shallower Fe-rich mantle is more ubiquitous for the KREEP-free basalts than the deeper mantle source of PKT-region basalts. These results are also

supported by recent remote sensing observations of pyroxene-rich mantle[40,41] at shallow depth (~100 km)[40]. Indeed, the high proportion of pyroxene in the KREEP-free basalt mantle source(s) suggests mantle with pyroxene > olivine (Supplementary Fig. 11, Supplementary Text 5 and 8).

## A possible melting mechanism for KREEP-free basalts

The melting regimes in most terrestrial bodies, including Earth, Mars, Venus, and Mercury suggest the dominance of partial melting either by decompression of adiabatically rising melts, from thermally anomalous, hot and buoyant portions of the mantle, or by lowering the solidus with addition of $H_2O$-dominated volatiles (flux) into the mantle. In contrast, it has been widely accepted that lunar mantle melting was aided by heating from radioactive elements in KREEP materials to generate mare basalts. Our calculation shows that the primary radioactive elements (K, Th, and U) can produce heat in the order of ~10⁹ J/g/y for the terrestrial planets, corresponding to 1 °C increase of temperature over ~10⁷ years for 1 g of peridotite. Considering an end-member case of extreme (100%) enrichment of the urKREEP layer in the Moon's mantle, 1 °C temperature rise in 1 g peridotite can take place over a period of ~10⁵ years. In the previously studied lunar returned samples and meteorites, the maximum KREEP-enrichment observed is <40% in a few lunar samples and meteorites[5,42] (e.g., 14310, 15382, 15386, SaU 169). If we assume this maximum case as the enrichment of urKREEP layer in the Moon's mantle, then the increase of temperature by 1 °C could have taken place between 10⁷ and 10⁵ years.

To melt the Moon's mantle, however, the supplied heat by radioactive source must also cross the latent heat of fusion of minerals and rocks, which is few hundred times more than the specific heat required to bring the minerals and rocks up to their melting temperatures. Further, dissipation of heat or thermal decay would hinder the melting process. This makes the production of sufficient melt even more difficult. Furthermore, being incompatible, the heat-producing

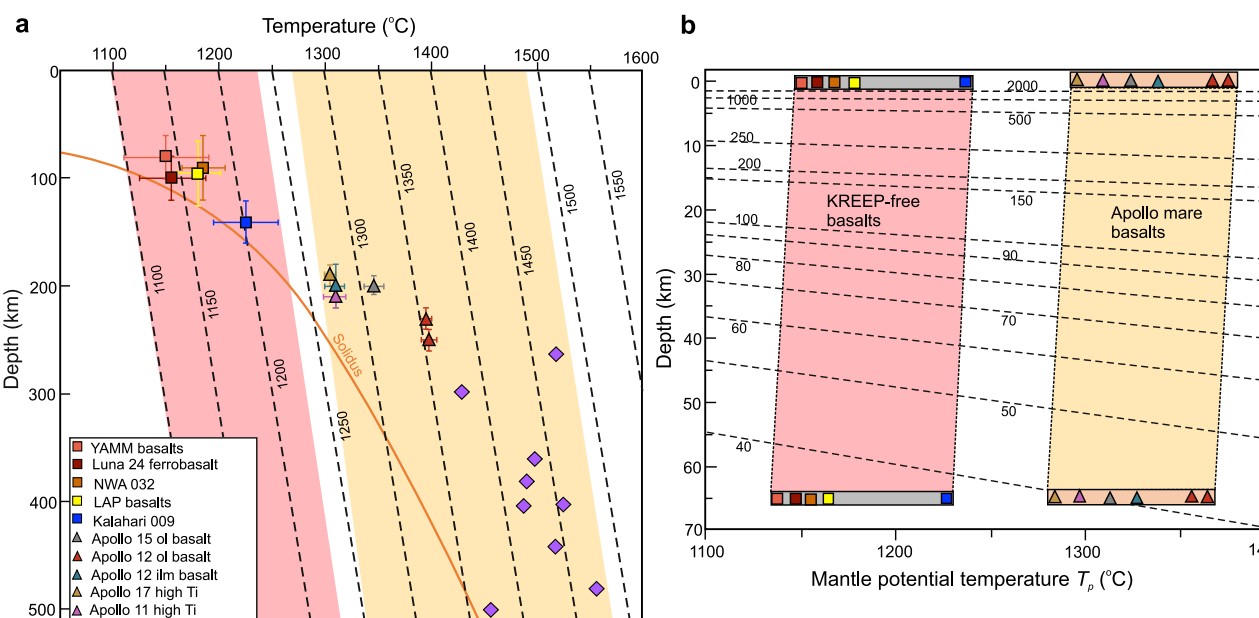

**Fig. 4 | Formation temperature ($T$), pressure ($P$) and surface heat flow of KREEP-free basalts compared to other lunar lithologies. a** The KREEP-free basalts (red field) show lower formation $P$-$T$ (see Methods and Supplementary Data 4) than Apollo mare basalts and pyroclastic glasses (orange field). Mantle potential temperatures ($T_p$) in °C, are shown as dashed lines. The estimates of $P$-$T$ for pyroclastic glasses are multiple saturation point data and are adopted from refs. 31, 36 and references therein. The $P$-$T$ for the KREEP-free and Apollo mare basalts are obtained

from pMELTS and thermobarometry (Methods, Supplementary Text 6 and 7). Solidus curve is drawn based on refs. 37, 48. **b** Surface heat flow is calculated assuming variable depth (0–65 km) of magma stalling. For a constant assumed depth, KREEP-free basalts show lower estimated heat flow (shown as dashed lines in mW/m²) than Apollo mare basalts (Methods and Supplementary Data 4). Lunar pressure-depth relationship is from ref. 48.

elements would largely concentrate in the melt and escape, leaving the residual mantle too depleted to melt further and produce the volcanism observed at the lunar surface[43].

An estimated total mare basaltic fill volume on the Moon is ~$8.6 \times 10^6$ km$^3$ (ref. 44). Here, we calculate the energy stored in the 10 km KREEP-rich layer and evaluate whether the energy budget was enough to generate the mare basalts on the lunar surface as a first order approximation. Assuming the density as 3270–3460 kg/m$^3$ and the latent heat of crystallization as $4 \times 10^5$ J/kg of basalts, our first order estimate indicates that the total energy required to produce ~$2.9 \times 10^{10}$ kg mare basalts is ~$1.2 \times 10^{25}$ J (or $3.3 \times 10^9$ TWh). In contrast, the total energy obtained from the radiogenic heat, for an assumed 10 km thick global urKREEP layer with a density close to ilmenite bearing cumulates (~3719 kg/m$^3$), is only $5.49 \times 10^7$ TWh, which is not a sufficient (<2%) total energy budget. The radioactive materials are concentrated within the PKT region, while the calculated volume of mare volcanism is probably more substantial as some of the older volcanic signatures are buried as cryptomare. Radioactive heating might have played an important role in generating the PKT KREEP basalts but the generally limited KREEP component seen in most Apollo mare basalts suggest that the energy generated by radioactive materials alone cannot be a major driving force for lunar magmatism. This contrasts with the earlier studies[39], which postulated radioactive and internal heating as the driving mechanisms for melting in the PKT terrain assuming a completely molten (KREEP) layer between the lunar crust and mantle, globally. This model of mare basalt generation is strictly valid for KREEP-rich samples (residing in the PKT region) with unusually high amounts of KREEP addition, which have been questioned on the basis of the variable depth of mare basalt generation and the viability of heating mechanisms, e.g., adiabatic decompression versus conductive heating, to form a small amount of melt generated at variable depths in the mantle source regions[36,45]. Alternatively, impact-induced decompression might have also engendered melting of the lunar mantle, given the coincidence of YAMM basalts' crystallization ages with the LHB.

A perhaps more likely scenario is that the latent heat of lunar formation together with the interior heat sources were sufficient to engender low degree partial melting of pre-existing mafic LMO cumulates, at least until the youngest Chang'E 5 basalt volcanism (~2.0 Ga)[46]. Our result shows that the formation mechanism of KREEP-free basalt volcanism cannot solely due to heating by radioactive elements. Nevertheless, the oldest KREEP-free basalts, those originated from the shallowest depths, most probably underwent low-pressure decompression type melting pyroxene-rich mantle in contrast to the traditional idea that they were the byproducts of lunar mantle overturn.

## Methods

### Scanning electron microscope analysis

The studied samples were provided by the National Institute for Polar Research, Japan (NIPR). Analysis was performed on two polished thick sections of A-881757 (88a, 88b). Detailed imaging of the sections was carried out on a JEOL JSM-7100F field emission scanning electron microscope (FE-SEM) equipped with an energy dispersive spectrometer (Oxford AZtec Energy) at the NIPR. The sections were coated with carbon. Acceleration voltage was 15 kV. Back-scattered electron (BSE) and X-ray mapping were taken (Supplementary Figs. 1 and 2). The X-ray images obtained for different elements were merged together using AZtec energy software, for the combined elemental X-ray image.

### Electron probe microanalysis

Major and minor-element compositions of minerals were determined using JEOL JXA-8200 electron probe micro-analyzers (EPMA) at the NIPR. Olivine, chromite, ilmenite, troilite, and Fe-metals were analyzed with a beam current of 30 nA by a focused beam, plagioclase (maskelynite) with a current of 10 nA by a focused beam, and phosphates with a beam current of 5 nA by a broad beam (~5 µm in diameter), all at 15 kV. Data were reduced using a ZAF correction procedure. The typical counting time for all elements in minerals was 30 s except for Na. The Si, Mg, Fe Mn, Ca, Na, P, and Al were analyzed using PET, PETH, LIF, LIFH, and LDE spectrometer crystals. The standards used for elements are natural and synthetic materials. The results are given in Supplementary Data 1.

### Inductively coupled plasma mass spectrometry

Given the coarseness of constituent primary mineral phases in A-881757 and the limited masses typically available for study, a single reported analysis may not be truly representative of the average bulk composition of the sample. The measurement of a representative bulk chemical composition for A-881757 required a substantial amount of the sample (0.71 g) to be homogenized and analyzed for a texturally and mineralogically representative sample with no sampling bias.

Analytical procedures were undertaken at the Scripps Isotope Geochemistry Laboratory (SIGL) and Physical Research Laboratory (PRL). For major and trace element abundances, A ~50 mg aliquot of homogenized sample powder was digested in Teflon-distilled concentrated HF (4 mL) and HNO$_3$ (1 mL) for >72 h on a hotplate at 150 °C, along with total procedural blanks and terrestrial basalt and andesite standards (BHVO-2, AGV-2). Samples were sequentially dried and taken up in concentrated HNO$_3$ to destroy fluorides, followed by doping with indium to monitor instrumental drift during analysis, and then diluted to a factor of 50,000 for major-element determination and 5000 for trace-element determination. Major-element abundances were obtained using a ThermoScientific iCAP Qc quadrupole inductively coupled plasma mass spectrometer (ICP-MS) in low resolution mode. For major-elements, Si was derived by difference, with reproducibility of other elements measured on the BHVO-2 reference material being better than 3%, except Na$_2$O (7.1%). The results are provided in Supplementary Data 2.

### Mantle melting model

We applied both batch and fractional melting to calculate the REE concentrations in the parental melts to ascertain the petrological history of the samples, assuming that each mineral phase melts in proportion to its modal abundance in the source. In most cases, the bulk rock composition was taken as a parental melt composition for the reasons discussed in Supplementary Text 3. Assuming the parental melt compositions represent compositions of the primary melt and no major process had occurred to significantly change the REE composition between source partial melting and parental melts, we calculated the degree of partial melting that likely occurred to generate each rock. The chemical modeling, similar to Snyder et al.[25] and Hallis et al.[5], was applied assuming an initial LMO source REE composition of 3× chondrite REE composition, using the rationale of Hughes et al.[47]. With the advancement of LMO crystallization, equilibrium crystallization gave way to fractional crystallization at about 50–70 PCS[25,26,48–53]. Plagioclase appeared on the liquidus just after ~75 PCS in the modeling, similar to experimental and thermodynamic modeling results[52,53]. Given YAMM, Kalahari 009 and Luna 24 ferrobasalts were low-Ti varieties with low REE abundance and had slight negative to positive Eu anomaly, the best fit was obtained by assuming a modal mineralogy close to 75–80 PCS LMO crystallization with addition of 1–2 % TIRL (Supplementary Fig. 9 and Supplementary Data 3). While for NWA 032 and LAP basalts, our estimated model mineralogy was obtained at 86 PCS with addition of 1–2 % TIRL (Supplementary Figs. 9 and 14, Supplementary Data 3). All deduced source modal mineralogy shows a dominance of clinopyroxene (pigeonite) over early formed olivine and orthopyroxene with small amount of plagioclase suggesting that KREEP-free basalts were sourced from a pyroxene-rich mantle. The individual source mineralogy of the studied samples and their feasibility is discussed in Supplementary Text 3 and 5. The REE

partition coefficients for olivine[54], orthopyroxene[55], augite[55], pigeonite[56] and plagioclase[57], are shown in Supplementary Table 1.

Batch and fractional melting are two end member process and in nature partial melting can be a combination of the two. Batch melting is calculated using the following equation:

$$C_L/C_0 = 1/(D_0 + F(1 - D_0)) \tag{1}$$

while fractional melting is calculated using equation:

$$C_L/C_0 = (1/D_0) \times (1 - F)^{\left(\frac{1}{D_0} - 1\right)} \tag{2}$$

where $C_L$ is the weight concentration of a trace element in the melt, $C_O$ is the weight concentration of a trace element in the original cumulate source, $F$ is the weight fraction of melt produced and $D_O$ is the bulk distribution coefficient of the original solid material. The bulk distribution coefficient is calculated by multiplying each mineral partition coefficient with the fraction of that mineral in the source.

### Estimation of pressure (P) and temperature (T)
We applied two methods to estimate the pressure and temperature condition of the studied rocks. First, we used pMELTS model calculations. The pMELTS mode[32] of program alphaMELTS, which is based on equilibrium phase diagram calculations and uses an internally consistent thermodynamic data set[33,58], was used to calculate formation $T$ and $P$ of A-881757, other YAMM and non-KREEP basalts. Since A-881757 and other YAMM meteorites are not extensive fractionation products and rather represent parental melt composition from their mantle source, an equilibrium batch crystallization is more suitable than the fractionation process. Therefore, the equilibrium phase diagram calculation method may infer actual $P$–$T$ conditions. Previous studies have shown that these phase diagram calculations are capable of providing models and comparing that with petrographic observations in the terrestrial[32,58], martian[59–61] and lunar[30,62–64] mantle and melt. The phase diagram calculations further allow the modes and composition of the mineral and melt phases to be tracked through $P$–$T$ space, which along with the aid of petrographic observations can yield the $P$–$T$ conditions of crystallization. A range of pressure (12.0–0.001 kbar), temperature (1500–1000 °C), and $fO_2$ (IW −1.0) conditions were set for the crystallization calculations in this method.

Second, we used thermobarometry techniques. With no signs of extensive fractional crystallization, the composition of the most Mg-rich minerals (high Mg#) were considered to be in equilibrium with the parental melt and were chosen for the thermobarometric calculations. The primary and the first crystallization phase in A-881757 and other YAMM meteorites is clinopyroxene. We used clinopyroxene-liquid[34], clinopyroxene-only[35,37] thermobarometry and olivine-liquid[65] thermometry to estimate the formation pressure and temperature of the YAMM basalts and other studied rocks. The list of thermobarometers along with the standard error of estimate (SEE) associated with each are compiled in Putirka[34]. In the case of a sample with olivine on their liquidus, we also performed olivine-liquid thermometry using the Beattie's[65] thermometer, which uses a liquid (whole rock) composition. This model is preferred over others because of its insensitivity to the $fO_2$ and the lowest SEE of ±44 °C. For calculations, we used the range of pressures estimated from phase diagram mode pMELTS, as well as clinopyroxene thermobarometers.

For YAMM meteorites which lack olivine, we applied clinopyroxene-liquid and clinopyroxene-only thermobarometry. We utilized equation 30 (for $P$) and 34 (for $T$) from Putirka[34] for clinopyroxene-liquid thermobarometry. The SEE for the pressure

and temperature are ±3.6 kbar and ±45 °C, respectively. The obtained results for the clinopyroxene-liquid thermobarometry are affected by volatile (Na) loss due to the open furnace nature of earlier experiments leading to systematically high $P$–$T$ estimates[66]. Clinopyroxene-only thermobarometers, however, obviate this problem. We used equations 32a (for $P$) and 32d (for $T$) from Putirka[34], and Eq. 1 (for $P$) and 2 (for $T$) from ref. 35 for clinopyroxene-only thermobarometry. The given SEE for the equations 32a and 32d is ±3.1 kbar and ±58 °C, respectively. Wang et al.[35] provided a new clinopyroxene-only thermobarometer, which is applicable to most basaltic-andesitic compositions and is insensitive to the fugacity conditions (Fig. 8 of Wang et al.[35]) with ±36.6 °C and ±1.66 kbar SEE. Further we compare the range of pressure and temperature obtained from both the methods in Supplementary Text 4 and Supplementary Data 4.

### Estimation of mantle potential temperature
The estimated pressure and temperature of formation of the YAMM basalts allow us to place further constraints on the mantle potential temperature for the Moon and its plausible evolution following the approach of Filberto and Dasgupta[67,68]. The calculation was performed first by evaluating the percentage of melt fraction needed to produce the primary magma composition from trace element modeling (Supplementary Text 3). Further, to calculate the mantle potential temperature, $T_p$, of the lunar mantle, we also corrected the average pressure and temperature of the bulk Moon composition to zero pressure using a lunar mantle adiabatic gradient of 0.17 K/km[36]. We corrected the effect of latent heat of fusion on the temperature using the expression:

$$\triangle T_{fus} = F\left(\frac{H_{fus}}{C_p}\right) \tag{3}$$

where $F$ is the melt fraction; $H_{fus}$ ($6 \times 10^5$ J K$^{-1}$kg$^{-1}$; ref. 69) is the heat of fusion; and $C_p$ (1000 J K$^{-1}$kg$^{-1}$; ref. 69) is the heat capacity at a constant pressure. Finally, we use the formula:

$$T_p = T_{avg\,eq} + \triangle T_{fus} - \triangle T_{lg} \tag{4}$$

where $T_{avg\_eq}$ is the average equilibrium (formation) temperature; $\triangle T_{fus}$ is the latent heat of fusion; and $\triangle T_{lg}$ is the lunar temperature gradient correction for adiabatic cooling.

### Estimation of surface heat flow
The thermal boundary layer structure of a terrestrial body mainly comprises of two parts; in the upper part (crust and upper mantle), heat is transported by conduction while convection dominates in the lower part (lower mantle), also called the convective boundary layer. In a steady state scenario and in the absence of heat-producing elements, heat flow is supposed to be constant in the conductive crust, suggesting a relatively constant temperature gradient for constant thermal conductivity. The calculated steady state heat flux at the base of the lithosphere may directly represent the surface heat flux (assuming no contribution of heat sources in crust and lithospheric mantle). Therefore, in the absence of heat-producing elements, the heat flow is calculated using the relationship between the depth and the mantle potential temperature[70]. We use here the steady state equation:

$$Q = k_{crust}\left(\frac{T_p - T_s}{d}\right), \tag{5}$$

where $Q$ is the heat flow at the surface; $k_{crust}$ is the thermal conductivity of lunar crust; $T_p$ is the mantle potential temperature; $T_s$ is the

surface temperature; and $d$ is the depth of the conductive (crust) - convective (mantle) boundary layer that is the thickness of crust. Considering the persisting uncertainty in the lunar thermal boundary layer structure and its rheological properties, we took a simplistic case where we calculated the heat flow assuming a constant thermal conductivity of crust, $k_{crust}$ -2.0 W m$^{-1}$ K$^{-1}$ (ref. 71) and variable thickness of conductive boundary layer (0–60 km), consistent with the GRAIL observations. The variable crustal thickness allows us to compare the thermal condition during the placement of the non-KREEP basalts and Apollo mare basalts, independent to their age and location (Fig. 4b and Supplementary Fig. 13). Heat flow values are reported for varying depth in Supplementary Data 4.

## Data availability
The data generated in this study are provided in the Supplementary files.

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

## Acknowledgements

Y.S. and A.B.S. acknowledge the support by the Indian Space Research Organization, Department of Space, Government of India. This work is a part of the PhD work of Y.S. Director PRL, Head of Planetary Science Division, PRL, and Director IIT Gandhinagar are gratefully acknowledged for constant encouragement during the work. J.M.D.D. acknowledges the NASA Solar System Workings program for supporting his participation in this work (80NSSC22K0098). A.Y. acknowledges the support from JSPS KAKENHI (JP19H01959) and NIPR Project Research (KP307). A.T. acknowledges the support from JSPS KAKENHI (JP19J00954).

## Author contributions

Y.S. and A.B.S. conceived the research and developed the central ideas. A.B.S. and J.M.D.D. helped in refining the ideas. Y.S. and A.B.S. performed melting-crystallization model calculations, thermobarometric calculations, mantle potential temperature, and heat flow calculations. J.M.D.D., Y.S., and A.B.S. performed bulk analyses. A.Y. and A.T. performed in-situ analyses. Y.S. and A.B.S., analyzed the results, interpreted the data and wrote the manuscript with inputs from J.M.D.D., A.Y. and A.T.

## Competing interests

The authors declare no competing interests.
