## [Peer Review File · Nature Communications]

A changing thermal regime revealed from shallow to deep
basalt source melting in the MoonReviewer #1 (Remarks to the Author):

This study investigated the petrogenesis of the Asuka-881757 lunar meteorite (3.9Ga) and found that this KREEP-free basalt was derived from a cooler and shallower mantle source compared with the late-erupted Apollo mare basalts (3.8 to 3.3Ga), indicating a fundamental shift in melting regime in the Moon from ~3.9 to ~3.3 Ga. If the calculation method of the formation P-T is reliable, I think this paper is suitable for publication in Nature Communications because these results will be beneficial in understanding the volcanic and thermal history of the Moon. However, there are still some problems with the calculation of P-T. I do not recommend this paper to be accepted until these problems are solved,

First, the formation P-Ts of YAMM meteorites are estimated by pMELTS and pyroxene thermobarometric approaches. However, the formation P-Ts of the Apollo mare basalts are estimated by multiple saturation points of experimental results. Thus, the different P-T between YAMM meteorites and Apollo mare basalts may reflect the systematic bias of different methods. Since a large set of whole-rock and pyroxene chemical data of Apollo basalts has been reported, the formation P-T of the Apollo mare basalts can also be calculated by pMELTS and pyroxene thermobarometric approaches used in this study. Many Apollo basalts are similar in major elements to YAMM meteorites (e.g., low Mg# values, see Fig. 1). I cannot imagine their formation temperatures being 100-200 degrees higher than those of YAMM meteorites.

Second, this study assumes that the measured whole-rock compositions could reflect those of parental melt. The authors discussed this assumption and ruled out fractional crystallization and assimilation effect in the A-881757 meteorite. However, I think it is not enough. (1) YAMM meteorites exhibit signs of partial mineral (olivine and pyroxene) accumulation while olivine and pyroxene composition of NWA 032 and LAP basalts are in-equilibrium with the whole rock (Supplementary Fig. S5). The accumulation of early-crystallized olivine and pyroxene could result in the whole rock being more mafic than the parental melt, thus leading to a bias in estimating the formation P-T. (2) For NWA 032 and LAP, whether they have experienced fractional crystallization cannot be ruled out. Chemical modeling suggests that their REE can be reproduced by a small degree of partial melting. However, this modeling only provides one mechanism but cannot demonstrate that the small degree of partial melting is the only mechanism. Therefore, the authors need to provide additional evidence demonstrating that fractional crystallization must not occur. Otherwise, the calculation of the P-T using the whole-rock compositions of NWA 032 and LAP could be incorrect.

The third question is about the chemical modeling of REE (Extended Data Figure 2). The mantle sources of the A881757, Kalahari 009 and Luna 24 ferrobasalt are the same: 78PCS+1%TIRL. However, they exhibit different REE concentrations and patterns. Why? Does this mean that we can arbitrarily assume the mineral modes and REE compositions of the crystalline products of the lunar magma ocean (LMO)? I feel that the authors are just trying to match the final result. However, there are some problems. For example, the plagioclase has not yet crystallized at the 78PCS stage. Why could this source have up to 8% plagioclase? Where did this plagioclase come from? The late crystallized plagioclase was likely to float up. How did it largely get into the mantle source? By contrast, the 86PCS (mantle source of NWA 032 and LAP) could only contain 1% plagioclase, less than the 78PCS. I think the authors need to discuss why these different mineral modes are all feasible.

Forth, how did the Fe-rich mantle source form? The authors repeatedly emphasized that the YAMM meteorite was formed from a Fe-rich mantle source, but did not introduce the characteristics of the Fe-rich source and discuss how it was formed. Both 78PCS and 86PCS were considered to be the sources of Apollo basalts in previous studies. I think they are not Fe-rich; otherwise, the Apollo basalts should exhibit low Mg# either. An iron-rich endmember needs to be added to the mantle to form the Fe-rich source. Ilmenite-bearing cumulates? If so, the source should also be rich in titanium. However,

YAMM meteorites are all low-titanium basalt. How did this iron-rich mantle source form? The authors need to discuss the mechanism of its formation.

Fifth, lines 20-22, 32-35. This statement is incorrect. Many Apollo basalt samples are KREEP-free. See the Sr-Nd isotopes (Figure 2). The samples having $87\text{Rb}/88\text{Sr} < \text{Kalahari 009}$ and $147\text{Sm}/144\text{Nd} > \text{Kalahari 009}$ (the blue square) are all KREEP-free.

Sixth, Fig. 3. Illustrating the Apollo data using a single pattern for each site is misleading. Not all Apollo samples were enriched in LREEs as shown in this figure. There are several types of basalts in each site, which have distinct REE patterns. For example, A12 Ol, Pig, Ilm, A14 group A, B, C, A15 Ol, Pig. Many of them have depleted LREEs.

Reviewer #3 (Remarks to the Author):

Srivastava et al. investigate the geochemistry of ~ 3.9 by KREEP free basalts and via thermochemical modeling demonstrate that these rocks were formed from shallower and colder LMO sources than typical Apollo mare basalts and pyroclastic glasses. They conclude that the mantle potential temperature inferred from these basalts suggest a cooler mantle with lower surface heat flow compared to later mare basalts, and that this is indicative of a change in melting regime from these older shallower and cooler KREEP free sources to the later deeper hotter mare basalt sources where the basalts have KREEP signature.

Thermal evolution of the lunar mantle through its history is a poorly constrained aspect, and is key to understanding the temporal pattern of volcanism on the Moon and associated differentiation. In that respect, this study is bang-on in terms of significance to lunar science.

The authors have done a thorough job in the modeling and there were no glaring mistakes or omissions that were visible to me.

I do have comments that need to be addressed by the authors to clarify certain implications that they have stated. I recommend major revision and hopefully addressing these comments would improve the manuscript substantially.

-- Lines 28-31: Stating that a fundamental shift in melting regime took place during a certain time period is a strong statement and should be justified with a cause for temporal shift. If radioactive decay is not the reason to cause partial melting, what other processes could have caused the temporal change? Latent heat of LMO crystallization and interior heat sources or decompression melting - how would these cause a temporal change in melting regime? Latent heat and interior heat would likely decrease with time and decompression melting due to impacts would be tied to a period with high number of impacts (such as the LHB). But how would that explain older colder and shallower KREEP free basalts vs younger deeper KREEP bearing hotter ones? This needs to be clarified.

-- Lines 203-206 - I have to say - this is a very hand wavy statement based on assumptions that U, Th and K would be very incompatible ($D \ll 1$). The authors cite Turcotte and Ahern 1978 and there have been a couple studies since then, which have demonstrated that U in particular may not be as incompatible esp. in plagioclase (see Aigner-Torres et al., CMP 2007; DeVries et al., 2012 LPSC). Thus the authors' statement here has to be taken with a pinch of salt.

Lines 216-217: I understand the authors' argument against radioactive elements generating enough heat to partially melt the lunar mantle. However, this is in contrast to what Wieczorek and Phillips (2000, JGR-Planets) have demonstrated. A discussion is needed about why the two studies have contrasting outcomes - what may have been the differences in assumptions between the two studies that led to disparate conclusions?

Lines 219-220: References or calculation to support this?

Lines 223-226: Not clear - if latent heat of lunar formation (latent heat due to LMO crystallization) and interior heat sources (what sources - core cooling??) are sufficient to cause low degree partial melting, then why is impact induced decompression melting required for KREEP-free basalts?

**-- Ananya Mallik
University of Arizona**

Response to review comments- “A changing thermal regime revealed from shallow to deep basalt source melting in the Moon”

We are grateful and thankful to the reviewers for a thorough assessment of our manuscript and for providing us with constructive comments and suggestions. In the revised version, all the comments and suggestions have been taken into account and our responses are given in blue text under each comment.

Reviewer #1 (Remarks to the Author):

This study investigated the petrogenesis of the Asuka-881757 lunar meteorite (3.9 Ga) and found that this KREEP-free basalt was derived from a cooler and shallower mantle source compared with the late-erupted Apollo mare basalts (3.8 to 3.3 Ga), indicating a fundamental shift in melting regime in the Moon from ~3.9 to ~3.3 Ga. If the calculation method of the formation P-T is reliable, I think this paper is suitable for publication in Nature Communications because these results will be beneficial in understanding the volcanic and thermal history of the Moon. However, there are still some problems with the calculation of P-T. I do not recommend this paper to be accepted until these problems are solved,

First, the formation P-Ts of YAMM meteorites are estimated by pMELTS and pyroxene thermobarometric approaches. However, the formation P-Ts of the Apollo mare basalts are estimated by multiple saturation points of experimental results. Thus, the different P-T between YAMM meteorites and Apollo mare basalts may reflect the systematic bias of different methods. Since a large set of whole-rock and pyroxene chemical data of Apollo basalts has been reported, the formation P-T of the Apollo mare basalts can also be calculated by pMELTS and pyroxene thermobarometric approaches used in this study. Many Apollo basalts are similar in major elements to YAMM meteorites (e.g., low Mg# values, see Fig. 1). I cannot imagine their formation temperatures being 100-200 degrees higher than those of YAMM meteorites.

Response: *We agree with the reviewer’s comment that the difference could possibly reflect a systematic bias of different methods. To be internally consistent, we have now performed the*

*P-T calculation on some of the established parental melt composition of different Apollo sample suites (e.g., low-Ti 12002, 12020, 12016, 15555, and high-Ti 10050, and 74275). For example, the formation P-T obtained by pMELTS for 15555 is ~1.0 GPa and ~1350 °C, which is close to the results from MSP experiments (1.0-1.2 GPa and 1300-1350 °C, by **Kesson et al., 1975; Walker et al., 1977**). Since we are estimating similar P-T conditions, our obtained results mentioned in **Supplementary Table S4**, stand and indicate an actual persisting difference in the formation P-T of our studied samples and the Apollo mare basalts. However, to be consistent, we have made the requested changes in the **Supplementary Table S4** and discussed the selection of parent melt composition of Apollo mare basalts in **Supplementary Section S4**. We have added the calculated P-T by pMELTS and thermobarometry (wherever possible) of the newly selected parent melt compositions of Apollo mare basalts (e.g., low-Ti 12002, 12020, 12016, 15555, and high-Ti 10050, and 74275). We also have kept their MSP-derived P-T conditions in the table for comparison and discussed the obtained results in **Supplementary Section S6 and S7**.*

*Moreover, in support of our work, we present several recent studies that compare the thermodynamic models such as pMELTS and Perple_X calculations with the MSP experiments in details (**Elardo et al., 2021; Astudillo et al., 2022**). These studies have found their results close to the experimental value. These references were either already mentioned in the previous version (**line 524 of Main_text Methods**) or mentioned in the Supplementary Information (**line 439 of Section S7. Calculated P-T of the Apollo mare basalts**) of the Revised version. In fact, the time-consuming and expensive experiments show a single datum in P-T-X space while equilibrium thermodynamic models offer a mean to extrapolate experimental results seamlessly across various composition and P-T space.*

*We also agree with the reviewer that some of the low-Mg Apollo basalts may show similar low P-T conditions to the rocks of concern in this work but the fact that these samples are not likely to be parental melt compositions, restricts us to comment on the formation P-T of these samples. We can still predict that those low-Mg lunar basalts are rather fractionated from their parent melt composition. On the other hand, the studied samples especially YAMM meteorites are likely to reflect a parental melt composition (or close to it) as discussed in our **Supplementary section S3**.*

Second, this study assumes that the measured whole-rock compositions could reflect those of parental melt. The authors discussed this assumption and ruled out fractional crystallization and assimilation effect in the A-881757 meteorite. However, I think it is not enough. (1)

YAMM meteorites exhibit signs of partial mineral (olivine and pyroxene) accumulation while olivine and pyroxene composition of NWA 032 and LAP basalts are in-equilibrium with the whole rock (Supplementary Fig. S5). The accumulation of early-crystallized olivine and pyroxene could result in the whole rock being more mafic than the parental melt, thus leading to a bias in estimating the formation P-T. (2) For NWA 032 and LAP, whether they have experienced fractional crystallization cannot be ruled out. Chemical modelling suggests that their REE can be reproduced by a small degree of partial melting. However, this modelling only provides one mechanism but cannot demonstrate that the small degree of partial melting is the only mechanism. Therefore, the authors need to provide additional evidence demonstrating that fractional crystallization must not occur. Otherwise, the calculation of the P-T using the whole-rock compositions of NWA 032 and LAP could be incorrect.

***Response:** Based on trace element data, e.g., La/Sm versus La (Supplementary Fig. S6) and Ce/Yb versus Yb (newly added Supplementary Fig. S10), we suggest that A-881757 as well as MIL 05035 in the YAMM clan underwent partial melting, but did not experience extensive fractionation, and so represent a melt composition (Joy et al., 2008; Liu et al., 2009; this study). However, we also mentioned in the previous version of the manuscript that a certain percent of crystal accumulation is present (10-20%) within these rocks. While calculating the formation P-T, we considered this. We report the P-T using pMELTS considering the observed composition of coexisting pigeonite and augite in the studied section. The obtained result also corroborated with the clinopyroxene thermobarometry results. The estimated P-T for A-881757 (and YAMM) suggest a range of $P \sim 0.3-0.8$ GPa, and $T \sim 1100-1190$ °C (Supplementary Table S4).*

We completely agree with the reviewer's point about the various possibility of magmatic processes that NWA 032 and LAP basalts may have experienced, as previous studies (including some of our own) have suggested this (Richter et al., 2005; Day et al., 2006). As our work is more focussed on the older YAMM basalts than the Apollo mare basalts and the meteorites NWA 032 and LAP, we are focussing on the 3.9 Ga event. Because NWA 032 and LAP show similar KREEP-free (Sr-Nd isotope evidence) nature, that's why those are mentioned in this work. Therefore, we are discussing their petrogenetic history keeping in mind the underlying various possibilities of magmatic processes (controversies) more clearly in the present version (lines 116 to 133 of Main_text). Elardo et al., (2015) experiments on the evolved LAP and NWA 032 basalts show similar shallow formation P-T similar to our calculated P-T.

The third question is about the chemical modeling of REE (Extended Data Figure 2). The mantle sources of the A881757, Kalahari 009 and Luna 24 ferrobasalt are the same: 78PCS+1%TIRL. However, they exhibit different REE concentrations and patterns. Why? Does this mean that we can arbitrarily assume the mineral modes and REE compositions of the crystalline products of the lunar magma ocean (LMO)? I feel that the authors are just trying to match the final result. However, there are some problems. For example, the plagioclase has not yet crystallized at the 78PCS stage. Why could this source have up to 8% plagioclase? Where did this plagioclase come from? The late crystallized plagioclase was likely to float up. How did it largely get into the mantle source? By contrast, the 86PCS (mantle source of NWA 032 and LAP) could only contain 1% plagioclase, less than the 78PCS. I think the authors need to discuss why these different mineral modes are all feasible.

Response: *We agree with this comment and carefully checked the modal mineralogy of all the studied samples. A-881757, Kalahari 009 and Luna 24 ferrobasalts represent the same mantle source: 78PCS+1%TIRL, and they exhibit slightly different REE concentrations and patterns in the log scale. This is not arbitrarily arising from different mineral modes and REE compositions of the crystallized LMO at that stage of crystallization. It has been observed that size of the Eu-anomaly indicates the plagioclase variation in the source (Neal and Taylor, 1992; Snyder et al., 1992; Hallis et al., 2014). The samples A-881757, Kalahari 009 and Luna 24 ferrobasalts display a very small Eu-anomaly (either positive or negative or null) is because of certain amount of plagioclase in the source. In contrast, the Apollo mare basalts (e.g., low-Ti 12002, 12020, 15555, intermediate-Ti 12016, and high-Ti 10050, 74275) along with NWA 032 and LAP show large negative Eu-anomaly peaks, certainly indicating plagioclase undersaturated or very low plagioclase bearing mantle source (Neal and Taylor, 1992; Hallis et al., 2014). Therefore, we note that 78 PCS + 1% TIRL source of A-881757, Kalahari 009 and Luna 24 ferrobasalts with slightly variable modal abundances is capable of producing low variation in Eu-anomaly in the REE pattern among them. The REE patterns in these three samples have a distinctness in that they are all LREE depleted (viz., LREE/HREE < 1), unlike the Apollo mare basalts and NWA 032 and LAP basalts. In support of this, we noticed that the Eu-anomaly is slightly varying among the YAMM group itself (as shown in the **Extended Data Figure 2**). Since, there is a variation, therefore we like to provide a range i.e., 75-80 PCS that could the possibly fit the observed REE abundance and Eu anomaly (**Supplementary Fig. 11**).*

The plagioclase-bearing mantle source (up to ~8% plagioclase) has been previously suggested for other low-Ti/high-Al basalts such as 12038, 14321-type, Luna 16 and VLT Luna 24 basalts (Nyquist et al., 1981; Dickinson et al., 1985; Neal and Taylor, 1992). This shows that the amount of plagioclase varies from source to source and is primarily dependent on the observed Eu-anomaly and the REE pattern. In addition, the measured isotopic abundance in the studied samples also suggest the plagioclase-bearing mantle source (Supplementary Fig. S4). Based on measured high-Al (>11.5 wt.%; Supplementary Table S2) and very low REE abundance in Luna 24 ferrobasalt and Kalahari 009, such compositions would not be consistent with the assimilation of plagioclase rich crustal materials, as Eu abundance and Rb/Sr ratios would show considerable variations. This supports the notion that the aluminous nature of these basalts is a source feature.

In the model of LMO crystallization, plagioclase becomes saturated at ~75 PCS, consistent with experimental results of Rapp and Draper (2018) and modelling results of Johnson et al., (2021). The variable modal mineralogy of source and their plagioclase content emphasizes the inherent complexities in the lunar magma ocean and highlights the heterogeneity in the lunar mantle. Our primary aim from the trace elements modelling is to understand the degrees of partial melting required to generate these basalts. Thus, we model trace element evolution using the models that are akin to Snyder et al., (1992) and Hallis et al., (2014).

However, we have found a mistake in our model calculations. We thank the reviewer for pointing out the error in our fitted modal mineralogy. Although the modal mineralogy of olivine and orthopyroxene dominant cumulates matches with the observed trace element abundances, it would not correspond to a Fe-rich mantle. In this revision, we have modified the mantle mineralogy to be pigeonite dominant. These pigeonite dominant lithologies are likely to persist at 75-80 PCS and were also accompanied by plagioclase saturation in the LMO, within upper mantle of the moon (Johnson et al., 2021; Zong et al., 2022). This we discuss in more detail in the response of the next comment.

We incorporated the detailed information regarding feasibility of the modal mineralogy in the newly added Supplementary Information S5 (lines 302 to 366) and Supplementary Fig. S11. The Extended Data Figure 2, and Extended Data Table 1 are modified accordingly.

Forth, how did the Fe-rich mantle source form? The authors repeatedly emphasized that the YAMM meteorite was formed from a Fe-rich mantle source, but did not introduce the

characteristics of the Fe-rich source and discuss how it was formed. Both 78PCS and 86PCS were considered to be the sources of Apollo basalts in previous studies. I think they are not Fe-rich; otherwise, the Apollo basalts should exhibit low Mg# either. An iron-rich endmember needs to be added to the mantle to form the Fe-rich source. Ilmenite-bearing cumulates? If so, the source should also be rich in titanium. However, YAMM meteorites are all low-titanium basalt. How did this iron-rich mantle source form? The authors need to discuss the mechanism of its formation.

Response: *We thank the reviewer for the thorough insight, which is very helpful. We agree that the earlier modeled mantle mineralogy does not corroborate with the Fe-rich mantle source. In this revised manuscript, we have modified the source mineralogy that is pigeonite dominant and likely to be more Fe-rich than previously shown olivine and orthopyroxene dominant mineralogy. With the advancement of LMO crystallization, the equilibrium crystallization gives way to fractional crystallization at about 50-70 PCS (Snyder et al., 1992; Longhi 2003, 2006; Elkins-Tanton et al., 2011; Elardo et al., 2011; Charlier et al., 2018; Rapp and Draper, 2018; Johnson et al., 2021). This would have the effect of forming chemical layering in the upper lunar mantle, the deeper Mg-rich mantle to shallower Fe-rich mantle. Our study shows that the studied samples originated within the shallower mantle than the Apollo basalt melts (e.g., low-Ti 12002, 12020, 12016, 15555, and high-Ti 10050, and 74275). However, a smaller number of the Apollo basalts exhibit low Mg#, which has been explained to not be the melt composition, and they are the product of extensive fractionation (Snyder et al., 1990; Neal and Taylor, 1992; Snyder et al., 1992; Neal et al., 1994; Hallis et al., 2014). Therefore, the Fe-rich Apollo basalts not necessarily represent a shallow source, instead they are highly fractionated. However, their formation P-T condition is not retrievable. Since our studied samples are KREEP-free, we are not able to connect the ilmenite-bearing late cumulates overturn in their respective mantle source. Therefore, shallow Fe-rich mantle is the most probable mantle source for the low-titanium and REE-depleted YAMM meteorites.*

The modal mineralogy and its Fe-rich nature is also supported by experimental and modelling studies following fractional crystallization of TWM composition (Rapp and Draper, 2018; Johnson et al., 2021). In fact, our chosen modal source mineralogy also supports some of the remote sensing observation as well. Pyroxene rich mantle at shallow depth (~100km; Melosh et al., 2017) have been previously suggested by remote sensing observations within the SPA basin (Melosh et al., 2017; Moriarty et al., 2021).

We discuss the formation of the Fe-rich mantle source in the present version at the Supplementary Information S5 (lines 302 to 366).

Fifth, lines 20-22, 32-35. This statement is incorrect. Many Apollo basalt samples are KREEP-free. See the Sr-Nd isotopes (Figure 2). The samples having $87\text{Rb}/88\text{Sr} < \text{Kalahari 009}$ and $147\text{Sm}/144\text{Nd} > \text{Kalahari 009}$ (the blue square) are all KREEP-free.

Response: *We have corrected both the sentences and the changes are mentioned below.*

Line 20-23: *“Some of the Apollo mare basalts generally show evidence for the presence in their source of a late-stage radiogenic heat-producing incompatible element-rich layer, known for its enrichment in potassium, rare earth elements, and phosphorus, or ‘KREEP’.”*

Line 32-35: *“There is a broad consensus that most of the Apollo returned mare basalts contain variable quantities of material derived from a KREEP (potassium, rare earth element and phosphorus) component mixed into their mantle source(s), possibly due to mantle overturn after extensive lunar magma ocean (LMO) crystallization, widely known as ‘urKREEP’.”*

Sixth, Fig. 3. Illustrating the Apollo data using a single pattern for each site is misleading. Not all Apollo samples were enriched in LREEs as shown in this figure. There are several types of basalts in each site, which have distinct REE patterns. For example, A12 Ol, Fig. 11m, A14 group A, B, C, A15 Ol, Fig. Many of them have depleted LREEs.

Response: *Fixed. The changes are reflected in Fig. 3 of the present version. We have added the REE pattern for each distinct group as mentioned by the reviewer.*

Reviewer #3 (Remarks to the Author):

Srivastava et al. investigate the geochemistry of ~3.9 by KREEP free basalts and via thermochemical modeling demonstrate that these rocks were formed from shallower and colder LMO sources than typical Apollo mare basalts and pyroclastic glasses. They conclude that the mantle potential temperature inferred from these basalts suggest a cooler mantle with lower surface heat flow compared to later mare basalts, and that this is indicative of a change

in melting regime from these older shallower and cooler KREEP free sources to the later deeper hotter mare basalt sources where the basalts have KREEP signature.

Thermal evolution of the lunar mantle through its history is a poorly constrained aspect, and is key to understanding the temporal pattern of volcanism on the Moon and associated differentiation. In that respect, this study is bang-on in terms of significance to lunar science.

The authors have done a thorough job in the modeling and there were no glaring mistakes or omissions that were visible to me.

I do have comments that need to be addressed by the authors to clarify certain implications that they have stated. I recommend major revision and hopefully addressing these comments would improve the manuscript substantially.

-- Lines 28-31: Stating that a fundamental shift in melting regime took place during a certain time period is a strong statement and should be justified with a cause for temporal shift. If radioactive decay is not the reason to cause partial melting, what other processes could have caused the temporal change? Latent heat of LMO crystallization and interior heat sources or decompression melting - how would these cause a temporal change in melting regime? Latent heat and interior heat would likely decrease with time and decompression melting due to impacts would be tied to a period with high number of impacts (such as the LHB). But how would that explain older colder and shallower KREEP free basalts vs younger deeper KREEP bearing hotter ones? This needs to be clarified.

Response: *Mare basalt generation has mostly been understood in terms of Apollo mare basalts which cluster between 3.8 and 3.1 Ga, with peak at ~3.3 Ga (Stoffler et al., 2006; Hiesinger et al., 2011). The LHB event (~ 3.9 Ga; Bottke and Norman, 2017) stands as a potential divide between the pre-mare and mare volcanism (Hiesinger et al., 2011). The older Kalahari 009 and YAMM basalts (> 3.9 Ga) and the younger (3.8-3.1 Ga) Apollo basalts allow us to compare the two aspects of lunar volcanism across the chronological divide. While the younger basalts are traditionally believed to be the products of LHB + mantle overturn + radioactive heating induced melting (Shearer and Papike, 1999; Elkins-Tanton et al., 2011), the older basaltic magmatism as depicted by studied samples, and their source melting shouldn't be LHB initiated and are supposedly not due to radioactive heating as they are KREEP-free. Our*

*petrological modelling suggests that these samples were formed via low-pressure decompression type melting (**decompression but not impact induced**). With the handful of these older basalts, the best we can predict about their source melting is analogous from terrestrial (those which are not plate-tectonically driven) and the single plate martian settings. “Mantle plumes” (although we are not coining this term yet in this paper as a conservative approach, because of the lack of such samples) aided by the latent heat of lunar formation together with the interior heat sources seems most likely scenario for explaining the melting regime before the LHB and the fundamental shift after LHB. The low-pressure decompression / adiabatic decompression melting is also postulated for the early stages of Moon (**Christiansen et al., 2022**). We agree with the Reviewer that latent heat and interior heat would likely decrease with time. However, at the PKT region, the shift in melting regime could possibly be the result of cumulate mantle overturn (**Elkins-Tanton et al., 2011**), or sudden surge of impacts during ~3.9 Ga (**Elkins-Tanton et al., 2004**), or in combination of both, which were probably more prevalent than the latent heat and interior heat budget.*

*The KREEP-free Kalahari 009 and YAMM basalts are older than the Apollo basalts. Our study shows that the studied samples originated in the shallower mantle compared with Apollo basalt melts (e.g., low-Ti 12002, 12020, 15555, and high-Ti 10050, 12016, 74275). However, our study also supports a form of chemical layering in the upper mantle of the Moon. The deeper Mg-rich mantle to shallower Fe-rich mantle. There are samples in the Apollo basalts suites exhibit low Mg#, which has been explained to not be the melt composition, and they are the product of extensive fractionation (**Snyder et al., 1990; Neal and Taylor, 1992; Snyder et al., 1992; Neal et al., 1994; Hallis et al., 2014**). Therefore, the evolved Fe-rich Apollo basalts may not necessarily represent a shallow source, instead they are highly fractionated. Their formation P-T condition is also not retrievable because they are not a representative melt composition. The colder KREEP-free samples and the hotter Apollo samples are defined with respect to their mantle potential temperature, which depends on their respective source depth.*

-- Lines 203-206 - I have to say - this is a very hand wavy statement based on assumptions that U, Th and K would be very incompatible ($D \ll 1$). The authors cite Turcotte and Ahern 1978 and there have been a couple studies since then, which have demonstrated that U in particular may not be as incompatible esp. in plagioclase (see Aigner-Torres et al., CMP 2007; DeVries et al., 2012 LPSC). Thus the authors' statement here has to be taken with a pinch of salt.

Response: We agree with the reviewer regarding the pertaining uncertainty in the partitioning behaviour of elements in LMO melts and that U may behave slightly compatible in plagioclase rich melt. **De Vries et al. (2012)** measured the behaviour of U and other incompatible elements in pure plagioclase melt while **Aigner-Torres et al., (2007)** measured in the natural MORB composition. Incorporation of these results will affect the overall heat budget of the bulk Moon by decreasing the concentration of heat producing elements in the bulk Moon, as suggested by **De Vries et al. (2012)**.

However, our viewpoint was to address the general incompatible behaviour of these elements in the basaltic melt system. These elements as presented by the above study eventually act as incompatible elements and will preferentially incorporate in melts rather than solids. Also, our modelled mantle modal mineralogy shows very few percentages of the plagioclase in their source thus will not have significant effect on our interpretation.

However, we have made slight changes in our statement (**lines 215-217 of Main_text**) that “Furthermore, being incompatible, the heat-producing elements would largely concentrate in the melt and escape, leaving the residual mantle too depleted to melt further and produce the volcanism observed at the lunar surface.”

Lines 216-217: I understand the authors' argument against radioactive elements generating enough heat to partially melt the lunar mantle. However, this is in contrast to what **Wieczorek and Phillips (2000, JGR-Planets)** have demonstrated. A discussion is needed about why the two studies have contrasting outcomes - what may have been the differences in assumptions between the two studies that led to disparate conclusions?

Response: We have performed a first order calculation on the amount of energy that can be generated from the approximately 10 km global KREEP-rich layer alone and compare whether that is enough in generating the mare basalts that we observe on the lunar surface? Our results show that radiogenic heat to be insufficient in generating the mare basalts. We cannot directly compare our results to **Wieczorek and Phillip, (2000)** as we only calculated the energy stored in the 10 km KREEP-rich layer while **Wieczorek and Phillips, (2000)** have simulated melting in the PKT terrain assuming a completely molten layer (KREEP) between the lunar crust and mantle (taking into account both internal heat and radiogenic component). **Wieczorek and Phillip, (2000)**'s model of mare basalt generation is strictly valid for KREEP-rich samples (residing in the PKT region) with unusually high amounts of KREEP as mixing required to

achieve their results have not been observed in mare basalt samples. *Elkins-Tanton et al., (2003, 2004)* have questioned the existence of completely molten layer beneath the crust and suggested that “the existence of a completely molten zone is also inconsistent with the correlations of depths of origin with types of magma” and that “the small volumes of melt produced over a wide range of depths are more consistent with adiabatic decompression melting than with stationary conductive heating”. In addition, the melt generation in the model of *Wieczorek and Phillip, (2000)* is also dependent on early poorly constrained solidus-liquidus temperatures.

We have discussed above points in the lines 218-241 in the Main_text of the revised version.

Lines 219-220: References or calculation to support this?

Response: Thank you for this suggestion. In the absence of any previous study, it is very difficult to know whether the megaregoliths cover would be (or would not be) effective enough to retain the heat budget. Although it is true that the megaregolith cover has got limited extent over the Moon, but it will be difficult to prove based on any calculation, which is not the objective of this paper. Therefore, we removed this part from the text, which does not affect in anyway the results and interpretations of this work.

Lines 223-226: Not clear - if latent heat of lunar formation (latent heat due to LMO crystallization) and interior heat sources (what sources - core cooling??) are sufficient to cause low degree partial melting, then why is impact induced decompression melting required for KREEP-free basalts?

Response: We apologise for this confusion in the final conclusions of the manuscript. We wanted to point out that the KREEP-free basalts, specifically the older ones (Kalahari 009 and YAMM basalts) most possibly underwent the low-pressure decompression type melting rather than long believed idea of cumulate mantle overturn. We are not saying that this low-pressure decompression is explicitly caused by impacts (although it conceivably could be). The low-pressure decompression/adiabatic decompression melting is likely to occur in single plate planets such as Moon during early stages (*Christiansen et al., 2022*). This decompression on

the Moon could possibly be related to various styles of early mantle convection (e.g., density driven mantle overturn, and plumes?), thinning of lithosphere, and to the uplift of the mantle subsequent to formation of large impact basins (e.g., Elkins-Tanton et al., 2004).

We have modified this section with the new line numbers 234-249 in the Main_text of the revised version. First, we removed the Megaregolith part. We have elaborated the possibility of connection between our studied basalts and the KREEP-rich samples from the PKT region. Then, explored the possibility of impact-induced decompression as an alternative process. Finally, we ascertain a most likely scenario of the latent heat of lunar formation together with the interior heat sources are adequate for the melting of lunar interior for generation of the KREEP-free basalts by low-pressure decompression type melting.

References

- Aigner-Torres, M., Blundy, J., Ulmer, P. & Pettke, T. Laser ablation ICPMS study of trace element partitioning between plagioclase and basaltic melts: an experimental approach. *Contrib. to Mineral. Petrol.* 153, 647–667 (2007).
- Astudillo Manosalva, D. F. & Elardo, S. ~M. The Accuracy of Perple_X, pMelts, and MAGPOX in Modelling Equilibrium Crystallization of Lunar and Martian Basalt Compositions and Their Multiple Saturation Points. in *LPI Contributions* vol. 2678 2343 (2022).
- Bottke, W. F. & Norman, M. D. The late heavy bombardment. *Annu. Rev. Earth Planet. Sci.* 45, 619–647 (2017).
- Charlier, B., Grove, T. L., Namur, O. & Holtz, F. Crystallization of the lunar magma ocean and the primordial mantle-crust differentiation of the Moon. *Geochim. Cosmochim. Acta* **234**, 50–69 (2018).
- Christiansen, E. H., Best, M. G. & Radebaugh, J. The origin of magma on planetary bodies. in *Planetary Volcanism Across the Solar System* 235–270 (Elsevier, 2022).
- Day, J. M. D. *et al.* Comparative petrology, geochemistry, and petrogenesis of evolved, low-Ti lunar mare basalt meteorites from the LaPaz Icefield, Antarctica. *Geochim. Cosmochim. Acta* **70**, 1581–1600 (2006). **Elardo et al., 2015**

- De Vries, J., van Westrenen, W. & van den Berg, A. Radiogenic heat production in the Moon: constraints from plagioclase-melt trace element partitioning experiments. in *Lunar and Planetary Science Conference 1737* (2012).
- Dickinson, T. *et al.* Apollo 14 aluminous mare basalts and their possible relationship to KREEP. *J. Geophys. Res. Solid Earth* **90**, C365–C374 (1985).
- Elardo, S. M. & Astudillo Manoslava, D. F. Ancient Igneous Differentiation Trends in the Moon's Crust Can Be Produced by Secondary Magmatism from a Common Source. in *52nd Lunar and Planetary Science Conference* 2313 (2021).
- Elardo, S. M., Draper, D. S. & Shearer Jr, C. K. Lunar Magma Ocean crystallization revisited: Bulk composition, early cumulate mineralogy, and the source regions of the highlands Mg-suite. *Geochim. Cosmochim. Acta* **75**, 3024–3045 (2011).
- Elkins-Tanton, L. T., Burgess, S. & Yin, Q.-Z. The lunar magma ocean: Reconciling the solidification process with lunar petrology and geochronology. *Earth Planet. Sci. Lett.* **304**, 326–336 (2011).
- Elkins-Tanton, L. T., Hager, B. H. & Grove, T. L. Magmatic effects of the lunar late heavy bombardment. *Earth Planet. Sci. Lett.* **222**, 17–27 (2004).
- Hallis, L. J., Anand, M. & Strekopytov, S. Trace-element modelling of mare basalt parental melts: Implications for a heterogeneous lunar mantle. *Geochim. Cosmochim. Acta* **134**, 289–316 (2014).
- Hiesinger, H., Head, J. W., Wolf, U., Jaumann, R. & Neukum, G. Ages and stratigraphy of lunar mare basalts: A synthesis. *Recent Adv. Curr. Res. issues lunar Stratigr.* **477**, 1–51 (2011).
- Johnson, T. E., Morrissey, L. J., Nemchin, A. A., Gardiner, N. J. & Snape, J. F. The phases of the Moon: Modelling crystallisation of the lunar magma ocean through equilibrium thermodynamics. *Earth Planet. Sci. Lett.* **556**, 116721 (2021).
- Joy, K. H. *et al.* The petrology and geochemistry of Miller Range 05035: A new lunar gabbroic meteorite. *Geochim. Cosmochim. Acta* **72**, 3822–3844 (2008).

- Kesson, S. E. Mare basalts: melting experiments and petrogenetic interpretations. *Lunar Planet. Sci. Conf. Proc.* 1, 921–944 (1975).
- Liu, Y., Floss, C., Day, J. M. D., Hill, E. & Taylor, L. A. Petrogenesis of lunar mare basalt meteorite Miller Range 05035. *Meteorit. Planet. Sci.* **44**, 261–284 (2009).
- Longhi, J. A new view of lunar ferroan anorthosites: Postmagma ocean petrogenesis. *J. Geophys. Res. Planets* **108**, (2003).
- Longhi, J. Petrogenesis of picritic mare magmas: constraints on the extent of early lunar differentiation. *Geochim. Cosmochim. Acta* 70, 5919–5934 (2006).
- Melosh, H. J. *et al.* South Pole–Aitken basin ejecta reveal the Moon’s upper mantle. *Geology* **45**, 1063–1066 (2017).
- Moriarty, D. P., Dygert, N., Valencia, S. N., Watkins, R. N. & Petro, N. E. The search for lunar mantle rocks exposed on the surface of the Moon. *Nat. Commun.* **12**, 1–11 (2021).
- Neal, C. R. & Taylor, L. A. Petrogenesis of mare basalts: A record of lunar volcanism. *Geochim. Cosmochim. Acta* **56**, 2177–2211 (1992).
- Neal, C. R. *et al.* Basalt generation at the Apollo 12 site, Part 1: New data, classification, and re-evaluation. *Meteoritics* **29**, 334–348 (1994).
- Nyquist, L. E., Wooden, J. L., Shih, C.-Y., Wiesmann, H. & Bansal, B. M. Isotopic and REE studies of lunar basalt 12038: Implications for petrogenesis of aluminous mare basalts. *Earth Planet. Sci. Lett.* 55, 335–355 (1981).
- Rapp, J. F. & Draper, D. S. Fractional crystallization of the lunar magma ocean: Updating the dominant paradigm. *Meteorit. Planet. Sci.* **53**, 1432–1455 (2018).
- Righter, K., Collins, S. J. & Brandon, A. D. Mineralogy and petrology of the LaPaz Icefield lunar mare basaltic meteorites. *Meteorit. Planet. Sci.* **40**, 1703–1722 (2005).
- Shearer, C. K. & Papike, J. J. Magmatic evolution of the Moon. *Am. Mineral.* 84, 1469–1494 (1999).

- Snyder, G. A., Taylor, L. A. & Neal, C. R. A chemical model for generating the sources of mare basalts: Combined equilibrium and fractional crystallization of the lunar magmasphere. *Geochim. Cosmochim. Acta* **56**, 3809–3823 (1992).
- Snyder, G. A., Taylor, L. A. & Neal, C. R. The sources of mare basalts: A model involving lunar magma ocean crystallization, plagioclase flotation, and trapped instantaneous residual liquid. in *Mare Volcanism and Basalt Petrogenesis: Astounding Fundamental Concepts* 53 (1991).
- Stöffler, D. *et al.* Cratering history and lunar chronology. *Rev. Mineral. Geochemistry* **60**, 519–596 (2006).
- Tanton, L. T. E., Van Orman, J. A., Hager, B. H. & Grove, T. L. Re-examination of the lunar magma ocean cumulate overturn hypothesis: melting or mixing is required. *Earth Planet. Sci. Lett.* **196**, 239–249 (2002).
- Walker, D., Longhi, J., Stolper, E. M., Grove, T. L. & Hays, J. F. Slowly Cooled Microgabbros 15065 and 15555. in *Lunar and Planetary Science Conference* vol. 8 964 (1977).
- Wieczorek, M. A. & Phillips, R. J. The “Procellarum KREEP Terrane”: Implications for mare volcanism and lunar evolution. *J. Geophys. Res. Planets* **105**, 20417–20430 (2000).
- Zong, K. *et al.* Bulk compositions of the Chang’E-5 lunar soil: Insights into chemical homogeneity, exotic addition, and origin of landing site basalts. *Geochim. Cosmochim. Acta* (2022).

Reviewer #1 (Remarks to the Author):

The revision has taken into account the comments of reviewers and has been greatly improved. This version only leaves me with one question.

To calculate the P-T of a magma, whether we use pMELTS or pyroxene thermobarometers, what determines the temperature is the Mg# value. A magma with a high Mg# value has a high formation temperature, and a pyroxene with a high Mg# value has a high crystallization temperature.

As can be seen from Figure 1, KREEP-free basalts have lower Mg# values. It is not surprising that their formation temperatures are low. There are, however, many Apollo basalts that also have similar low Mg# values.

As explained by the authors, the low Mg# Apollo basalts have experienced various degrees of crystallization, whereas the KREEP-free basalts studied here have not been affected by crystallization, but instead were derived from Fe-rich mantle sources.

Due to this, only high Mg# Apollo basalts (12002, 12020, 12016, 15555, 10050, and 74275) are used when comparing formation temperatures. Obviously, KREEP-free basalts show low formation temperatures.

I feel that there is still a topic worth discussing in this paper. Are there any differences in petrology and geochemistry between the low Mg# Apollo basalts and the KREEP-free basalts?

As one is the result of extensive crystallization and the other is derived from pyroxenite mantle, there must be some significant differences between them. Based on knowledge of terrestrial basalts, basalts derived from pyroxenite mantle should have high Fe/Mn ratios and low Ni/Mg ratios (Sobolev et al., 2005, 2007). Therefore, I suggest that the authors make a comparison.

In addition, chemical modeling suggests that the mantle sources of KREEP-free basalts are not peridotite, but pyroxenite. The importance of such a conclusion needs to be highlighted in the main text rather than simply stating that it is a "pyroxene rich mantle".

References:

Sobolev, A.V., Hofmann, A.W., Sobolev, S.V. and Nikogosian, I.K. (2005) An olivine-free mantle source of Hawaiian shield basalts. *Nature* 434, 590-597.

Sobolev, A.V., Hofmann, A.W., Kuzmin, D.V., Yaxley, G.M., Arndt, N.T., Chung, S.L., Danyushevsky, L.V., Elliott, T., Frey, F.A., Garcia, M.O., Gurenko, A.A., Kamenetsky, V.S., Kerr, A.C., Krivolutskaya, N.A., Matvienkov, V.V., Nikogosian, I.K., Rocholl, A., Sigurdsson, I.A., Sushchevskaya, N.M. and Teklay, M. (2007) The amount of recycled crust in sources of mantle-derived melts. *Science* 316, 412-417.

Response to review comments- “A changing thermal regime revealed from shallow to deep basalt source melting in the Moon”

We are very grateful and thankful to the reviewers for their swift and thorough assessment of our manuscript, and for providing us with constructive comments and suggestions. In the revised version, all the comments and suggestions have been taken into account and our responses are given in blue text under each comment.

Reviewer #1 (Remarks to the Author):

The revision has taken into account the comments of reviewers and has been greatly improved. This version only leaves me with one question.

To calculate the P-T of a magma, whether we use pMELTS or pyroxene thermobarometers, what determines the temperature is the Mg# value. A magma with a high Mg# value has a high formation temperature, and a pyroxene with a high Mg# value has a high crystallization temperature.

As can be seen from Figure 1, KREEP-free basalts have lower Mg# values. It is not surprising that their formation temperatures are low. There are, however, many Apollo basalts that also have similar low Mg# values.

As explained by the authors, the low Mg# Apollo basalts have experienced various degrees of crystallization, whereas the KREEP-free basalts studied here have not been affected by crystallization, but instead were derived from Fe-rich mantle sources.

Due to this, only high Mg# Apollo basalts (12002, 12020, 12016, 15555, 10050, and 74275) are used when comparing formation temperatures. Obviously, KREEP-free basalts show low formation temperatures.

I feel that there is still a topic worth discussing in this paper. Are there any differences in petrology and geochemistry between the low Mg# Apollo basalts and the KREEP-free basalts?

As one is the result of extensive crystallization and the other is derived from pyroxenite mantle, there must be some significant differences between them. Based on knowledge of terrestrial basalts, basalts derived from pyroxenite mantle should have high Fe/Mn ratios and low Ni/Mg ratios (Sobolev et al., 2005, 2007). Therefore, I suggest that the authors make a comparison.

Response: *We thank reviewer for this insightful suggestion. We completely agree with the reviewer that the bulk Mg# of the rock has primary control on their formation temperatures, which is consistent with our results. However, there are low-Mg Apollo basalts with bulk Mg# similar to KREEP-free basalts and those have been suggested to have undergone varied degree of crystallization (Supplementary Section S4, S5). This difference could arise from the fact that low-Mg Apollo basalts are fractionally crystallized from their high-Mg counter-part which are thought to have come from an olivine rich mantle source in contrast to the pyroxene rich mantle source for KREEP-free basalts.*

As suggested by the reviewer, we explore the petrology and geochemistry of low-Mg Apollo basalts and KREEP-free basalts to highlight some of the variations that could arise due to difference in their source lithology. The melts from pyroxenite sources of the Earth are suggested to show higher Ni content at a given bulk MgO content when compared to their peridotite sourced counterpart (Sobolev et al., 2005; Sobolev et al., 2007; Herzberg, 2011) due to the retention of Ni in olivine. Several recent studies Yang and Zhou, (2013) and Yang et al., (2019) utilizes combination of ratio such as FC3MS value ($FeO/CaO - 3 * MgO/SiO_2$, all in wt.%) and FCKANTMS ($FCKANTMS = \ln(FeO/CaO) - 0.08 * \ln(K_2O/Al_2O_3) - 0.052 * \ln(TiO_2/Na_2O) - 0.036 * \ln(Na_2O/K_2O) * \ln(Na_2O/TiO_2) - 0.062 * (\ln(MgO/SiO_2))^3 - 0.641 * (\ln(MgO/SiO_2))^2 - 1.871 * \ln(MgO/SiO_2) - 1.473$, all the major elements in wt.%) to discriminate peridotite and pyroxenite sources.

We plot bulk composition of Ni, Al_2O_3 , FC3MS and FCKANTMS against bulk MgO (Supplementary Fig. S12). We could discern a crude set of clusters for the KREEP-free basalts from the low-Mg Apollo basalts, although the distinction is not that clear like the terrestrial counterparts. The plot (Supplementary Fig. S12) also shows a compositional gap between terrestrial and lunar basalts. This is most likely due to the difference in fO_2 and other formation conditions. Although, we observe similar variations in some terrestrial proposed markers (mainly in plots of FC3MS and FCKANTMS), we do not consider these variations to result from pyroxenite sources like those envisaged for Earth. This is because the initial Ni content of the bulk silicate Moon is different to that of the BSE, while the conditions of lunar mantle sources are also quite different to those in terrestrial settings (Day, 2020). As still there exists some controversy even in the terrestrial setting, but overall, as a pioneering work in case of the lunar basaltic magmatism this study may be useful in guiding future studies to propose markers for lunar basalt source identification. We further discuss the low-Mg Apollo basalt and KREEP-free basalt comparison in newly added Supplementary Section S8.

In addition, chemical modeling suggests that the mantle sources of KREEP-free basalts are not peridotite, but pyroxenite. The importance of such a conclusion needs to be highlighted in the main text rather than simply stating that it is a "pyroxene rich mantle".

Response: We agree that pyroxene-rich mantle is important, but we do not feel that pyroxenite is an appropriate terminology. For Earth, the mantle is dominantly peridotite, conforming well to seismic studies and mineral physics. For the Moon, magma ocean differentiation rendered a very different picture, and would have led to stratified olivine- and pyroxene-dominated sources. Calling them peridotite or pyroxenite goes too far and while a terrestrial comparison is tempting, such comparisons would be highly misleading at this stage.

References

- Sobolev, A.V., Hofmann, A.W., Sobolev, S.V. and Nikogosian, I.K., 2005. An olivine-free mantle source of Hawaiian shield basalts. *Nature*, 434(7033), pp.590-597.
- Sobolev, A.V., Hofmann, A.W., Kuzmin, D.V., Yaxley, G.M., Arndt, N.T., Chung, S.L., Danyushevsky, L.V., Elliott, T., Frey, F.A., Garcia, M.O. and Gurenko, A.A., 2007. The

amount of recycled crust in sources of mantle-derived melts. *Science*, 316(5823), pp.412-417.

Herzberg, C., 2011. Identification of source lithology in the Hawaiian and Canary Islands: Implications for origins. *Journal of Petrology*, 52(1), pp.113-146.

Yang, Z.F. and Zhou, J.H., 2013. Can we identify source lithology of basalt? *Scientific Reports*, 3(1), pp.1-7.

Yang, Z.F., Li, J., Jiang, Q.B., Xu, F., Guo, S.Y., Li, Y. and Zhang, J., 2019. Using major element log ratios to recognize compositional patterns of basalt: Implications for source lithological and compositional heterogeneities. *Journal of Geophysical Research: Solid Earth*, 124(4), pp.3458-3490.

Day, J.M.D., 2020. Metal grains in lunar rocks as indicators of igneous and impact processes. *Meteoritics & Planetary Science*, 55(8).